# Social Telepresence Robots: A Narrative Review of Experiments Involving Older Adults before and during the COVID-19 Pandemic

**DOI:** 10.3390/ijerph18073597

**Published:** 2021-03-30

**Authors:** Baptiste Isabet, Maribel Pino, Manon Lewis, Samuel Benveniste, Anne-Sophie Rigaud

**Affiliations:** 1EA 4468, Faculté de Médecine, Université de Paris, 75006 Paris, France; baptiste.isabet@brocalivinglab.org (B.I.); maribel.pino@aphp.fr (M.P.); samuel.benveniste@censtimco.org (S.B.); 2AP-HP, Hôpital Broca, 75013 Paris, France; 3School of Psychology, Faculty of Medical Sciences, Newcastle University, Newcastle upon Tyne NE1 7RU, UK; m.lewis6@newcastle.ac.uk; 4CEN Stimco, 75013 Paris, France

**Keywords:** older adults, telepresence robots, loneliness, COVID-19, health technology assessment

## Abstract

Social isolation is a common phenomenon among the elderly. Retirement, widowhood, and increased prevalence of chronic diseases in this age group lead to a decline in social relationships, which in turn has adverse consequences on health and well-being. The coronavirus COVID-19 crisis worsened this situation, raising interest for mobile telepresence robots (MTR) that would help create, maintain, and strengthen social relationships. MTR are tools equipped with a camera, monitor, microphone, and speaker, with a body on wheels that allows for remote-controlled and sometimes autonomous movement aiming to provide easy access to assistance and networking services. We conducted a narrative review of literature describing experimental studies of MTR involving elderly people over the last 20 years, including during the COVID-19 period. The aim of this review was to examine whether MTR use was beneficial for reducing loneliness and social isolation among older adults at home and in health and care institutions and to examine the current benefits and barriers to their use and implementation. We screened 1754 references and included 24 research papers focusing on the usability, acceptability, and effectiveness of MTR. News reports on MTR use during the COVID-19 period were also examined. A qualitative, multidimensional analysis methodology inspired by a health technology assessment model was used to identify facilitating and limiting factors and investigate if and how MTR could reduce social isolation in elderly people. Reviewed studies provide encouraging evidence that MTR have potential in this regard, as experiments report positive feedback on MTR design and core functionalities. However, our analysis also points to specific technical, ergonomic, and ethical challenges that remain to be solved, highlighting the need for further multidimensional research on the design and impact of MTR interventions for older adults and building on new insights gained during the COVID-19 pandemic.

## 1. Introduction

### 1.1. Social Isolation of the Elderly

Loneliness and social isolation among older adults (OAs) living alone in their own homes or in institutions is increasing. Social isolation is defined as the objective quantitative reduction in a person’s social network and number of contacts. It is usually distinguished from the feeling of loneliness, which lies more in a sense of dissatisfaction than in the absence of social relations [1].

In recent years, different countries have confirmed through national surveys that the prevalence of social isolation and of loneliness increases with age. In 2017, the International Federation of Little Brothers of the Poor, in association with Consumer Science and Analytics Research [2], conducted a survey on loneliness and isolation in people over 60 in France, estimated at 15 million in 2018 [3]. The survey revealed that 300,000 French people over the age of 60 were isolated from their social circle, some of them in extreme isolation. They showed that 22% of the elderly had no contact with their family circle, 28% with their circle of friends, 21% with their neighborhood circle, and 55% with their associative circle. Beyond the age of 85, the authors found that there was a breakdown of social circles, with significant reduction in social contacts and outings. Thus, 10% of the OAs in the 85–89-year-old bracket stayed at home for weeks on end without going out. This isolation is reinforced by the generational digital divide. According to the same study [2], 31% of people over 60 (68% of people over 85) are not familiar with the Internet. In Europe, it was estimated that almost 10% of people over 75 years of age are isolated in 2018 [4]. In the United States, 43% of the over 60s reported feeling lonely in 2020 [5]. In Canada, 12% of people over the age of 65 were said to be isolated in 2009 [6]. In Japan, it was estimated that 6 million elderly people will be living alone in 2016 [7]. Globally, these data show that loneliness and isolation are experienced by OAs all over the world.

This phenomenon of loneliness and isolation increased in the year 2020 when the coronavirus COVID-19 virus led to implementation of lockdowns, limiting social exchanges and physical contact, particularly among the elderly. The decrease in social contacts in this population was estimated to be between 75% and 90% during this period. Institutionalized older persons were particularly affected by the prohibition of visits and cancellation of group activities. Due to their lack of social ties, social exchanges, and interactions with relatives, care home residents experienced severe feelings of loneliness and abandonment [5,8].

Loneliness and isolation have particularly harmful consequences for psychological and somatic health. They can lead to the development of many disorders, such as depression [9], high blood pressure, poor quality of sleep, cognitive decline [10], chronic alcohol consumption [11], and an increased risk of mortality compared to non-isolated people [12,13].

### 1.2. Technologies That Help to Reduce Loneliness: Telepresence Robots

Several authors have called for the initiation of intervention strategies to alleviate loneliness and social isolation and improve OAs’ quality of life [14]. Various interventions have been proposed, including some that utilize technology such as computer with internet access, allowing for the use of messaging, social networks, chatrooms or forums, videoconferencing, and pet robots and conversational agents [15]. The authors concluded that these technologies were promising to reduce social isolation among seniors, but that further research was needed to evaluate the effectiveness of these technologies.

In 2020, the restriction of face-to-face contact due to COVID-19 created an opportunity for digital technology to be developed and used to reduce isolation and limit its impact on the health and quality of life of OAs. Thus, virtual visits have been set up in many healthcare establishments, and telepresence robots have been tested in various care structures [16,17].

Mobile telepresence robots (MTR) can be used to create a connection between two distinct environments and to set up social interactions between two individuals located in separate settings [18]. MTR are equipped with digital screens, cameras, microphones, and loudspeakers that allow remote interactions to be established, all of which are integrated into a platform that is itself connected to a system that enables movement (Figure 1). They have been developed for office work environments, schools, research, and healthcare [19,20].

MTR can also be used as social assistive robots for people with physical or cognitive disabilities [25]. Indeed, they can interact with users to promote their participation in activities such as communication, movement, domestic tasks, and health monitoring, and thus can improve their physical and psychological well-being [26]. All MTR provide telepresence thanks to the use of videoconference. Some of them can also include a set of services that facilitate the daily life of OAs, either by playing a compensatory role (reminders of appointments, tasks), providing cognitive stimulation (exercise programs), or increasing safety (calling of help providers or family in case of a fall).

MTR allow the user to be in contact with their social circle as well as healthcare professionals and to access leisure activities (cooking, yoga, or wellness workshops). They can be equipped with voice commands, allowing the user to make the robot perform tasks, such as moving around the home or informing them of appointments. MTR could induce a feeling of security for the OA and their carers, thanks to the possibility of quick and easy contacts. Use of video, for example, enables the carer to remotely accompany the OA in carrying out certain actions (e.g., indicating the right medication to take). In the event of a long no-contact period, the carer can take control of the robot remotely to navigate in the OA’s home and ensure that an incident, such as a fall, for example, has not occurred. MTR, which are used in the context of healthcare, can be considered as a health technology, defined as an intervention developed to prevent, diagnose, or treat medical conditions; promote health; provide rehabilitation; or organize healthcare delivery [27]. Under this definition, it is specified that an intervention can be a test, device, medicine, vaccine, procedure, program, or system.

Despite these potential benefits, MTR have not been well integrated into practice and penetrated care provision [28]. Moyle et al. [29] have highlighted several barriers (technical, organizational, sociological, and ethical) to the implementation and use of MTR in nursing homes. For instance, some users were reluctant to use these robots because of difficulties resulting from complex interfaces and lack of experience in manipulating them or fear of reducing real human interactions if the use of these tools was widespread. Overcoming these barriers is necessary in order to ensure proper adoption and implementation of these technologies into geriatrics care pathways.

The context of COVID-19-related social-life restrictions has sparked renewed interest in the use of social robots. Indeed, two authors have reported that MTR have enabled OAs to maintain contact with their social circle in order to obtain support, while at the same time contributing to protect care workers from infection [16,17]. These authors suggested that remote controlled systems could thus play an important role in the mental and physical well-being of people by engaging them socially, increasing the quality of their social interactions, and reducing the negative effects of social isolation. In addition, the COVID-19 virus pandemic that reduced face-to-face contacts between people could increase users’ perceived sense of usefulness of MTR and, consequently, the acceptance of their use.

So that MTR technologies can effectively contribute to good care and integrate healthcare service provision, it is important to consider several dimensions: clinical, technological, organizational, ethical, economic, etc. An analysis of how these dimensions have been addressed by research in the MTR field seems useful in order to identify for which dimensions scientific evidence exists and for which others a gap remains. One potentially suitable approach for this purpose is conducting a literature review of empirical work in the field using a multidimensional analysis framework such as those recommended by health technology assessment (HTA) models [30]. HTA provide methods and concepts, allowing for a global assessment of intended and unintended consequences of the use of a technology [31]. HTA is therefore used to inform the decision-making process concerning the introduction of new technologies to a health system. Considering the rapid grow of technological applications such as MTR to promote health and independence in elderly populations, this early HTA informed analysis could help the identification of critical areas for further research and development.

The aim of this review was to examine whether MTR use was beneficial for reducing loneliness and social isolation among OAs at home and in health and care institutions and to examine the current benefits and barriers to their use and implementation using a qualitative, multidimensional analysis methodology inspired by a HTA model. To our knowledge, this analysis has not been carried out yet.

## 2. Materials and Methods

### 2.1. Information Sources and Research Method

The research was conducted between May and September 2020. We reviewed publications published between January 2000 and September 2020 (i.e., from the past 20 years—including the period after the emergence of COVID-19). The objective was to identify publications discussing interventions using MTR at home and in health and care institutions for OAs, their families, and professionals. We examined users’ perspectives on the usability, feasibility, effectiveness, and acceptability of robotic interventions to identify facilitating and limiting factors. We used the multidimensional analysis framework recommended by health technology assessment models, which can clarify the ways for a good integration of these technologies in the process of provision of care [31].

Firstly, we consulted the following search engines: PubMed/Medline, Web of Science, Scopus, EMBASE, and PsychINFO. The keywords were grouped into two categories: *elderly* or *older* or *senior* or *elder*, and *telepresence robot* or *assistive robot* or *social robot*. The literature selected was done so on the basis of title, abstract, or full article. Secondly, secondary research using the internet and references from other articles was carried out according to the same inclusion criteria. We then searched using keywords grouped into 3 categories: elderly or older or senior or elder, telepresence robot or assistive robot or social robot, and COVID-19.

### 2.2. Criteria for Inclusion, Exclusion, and Data Extraction

We have included all publications in English or French in which OAs used MTR as part of an experimental intervention, regardless of the location (laboratory, hospital, institution, home) and regardless of the experimental design (observational study, case–control, randomized study, qualitative study). However, we did not include publications in which participants gave their opinion on the basis of photos or videos of robots without manipulating them. When several publications dealt with the same project, we only selected the publication giving the most detailed information about the work. The data were collected independently by 2 researchers (B.I. and A.-S.R.) after validating the extraction process on a small number of articles. The data collected about the experiments included the objectives, the characteristics of participants, the conditions of the experiments and evaluation tools, as well as the benefits and barriers to the implementation of the robots.

The flow chart summarizing the search and item selection strategy is shown in Figure 2.

We also classified the results using a multidimensional analysis grid, on the basis of the 9 dimensions of the European Network of Health Technology Assessment—EUnetHTA core model version 3.0 produced by the European Network of health technology assessment [32]. The model, publicly accessible, is provided for the production and sharing of HTA information, allowing for the support of evidence-based decision-making in healthcare. The HTA core model can be customized to the needs and objectives of the users as long as its defining characteristics are respected. Proper registration of the use of the model for this purpose was made on the EUnetHTA website [33]. The model comprises 9 critical dimensions of assessment (*domains*), each of which is subdivided in 2 sub-domains (*issues* and *topics*) to consider when assessing the use of health technologies. These dimensions are as follows: health issues and current use of technology, description and technological characteristics, safety, clinical effectiveness, cost and economic analysis, ethical analysis, organizational aspects, patient-focused and social aspects, and legal aspects (Table 1). For each dimension, we also presented our results at the topics/issues level of the EUnetHTA Core Model when possible.

## 3. Results

The analyzed references were published between 2003 and 2020. We did not find any scientific publications evaluating the impact of COVID-19 in a robotic telepresence experiment for elderly people. Selected relevant articles to the subject matter are presented in Table 2.

In the selected publications, robots were used in hospitals, laboratories, care homes for the elderly, and private homes. A total of 16 studies were carried out in Europe [26,35,36,38,39,40,41,42,43,44,45,46,48,50,53,54,55], 4 in America (USA) [19,34,49,51], 1 in Asia (Japan) [44], and 3 in Oceania (New Zealand and Australia) [37,47,52]. Experimental times ranged from 1 day to 18 months. The median was 2 days (see Table 2).

The selected articles included studies carried out with healthy OAs (four studies) [19,35,43,44], as well as OAs suffering from mild neurocognitive disorders (two studies) [36,49], major neurocognitive disorders or dementia (one study) [47], or multiple comorbidities (one study) [34]. Four studies included one healthy group and one mild cognitive impairment (MCI) group [19,34,49,51]. Twelve studies did not report information on the characteristics of the recruited OAs [26,38,40,42,45,47,48,49,50,52,55]. The number of older participants in the studies ranged from *n* = 1 to *n* = 53. Six studies included family members, [26,42,47,48,50,51], and eight included health professionals (nurses, orderlies, occupational therapists) [34,36,37,40,45,47,48].

The study design involved either cross-sectional studies including a small number of subjects who evaluated robots after a single session [19,35,36,39,40,41,43,45,46,49], after several sessions [38,44,50,51,53,54], or through the use of a longitudinal follow-up [19,34,38,43,48], as well as randomized studies [34,37]. The data collected was qualitative (focus groups, interviews), quantitative (completion of scales, recording of physiological phenomena), or mixed. These studies were very diverse and, as a result, not always easily comparable. Most of these studies included no comparators or baseline assessment. Some of the evaluations were very succinct, both qualitatively and quantitatively.

### 3.1. General Data

The synthesis of the results is presented in Table 3.

### 3.2. Description of Studies Using HTA Dimensions including Topics and Issues When Available

#### 3.2.1. Health Problem and Current Use of the Technology (CUR)

Target population for MTR included healthy OAs [19,35,43,44], as well as OAs suffering from mild neurocognitive disorders [36,49], major neurocognitive disorders or dementia [47], or multiple comorbidities [34].

##### Utilization of MTR

The MRT was used to increase communication between OAs and their environment through virtual calls and visits with the MTR. In all the studies, the objective was to evaluate the virtual interactions of an OA with either a family member, professional, or researcher participating in the study, facilitated by the MTR. Two publications [34,37] involved remote monitoring of the health status of OAs by professionals, while another [44] analyzed online education groups led by teachers. The objectives of the evaluations were to assess usability, acceptance, potential impact or objective benefits and limitations. Two publications dealt with psychological factors favoring acceptance of robots [35,52].

#### 3.2.2. Description and Technical Characteristics of Technology (TEC)

##### Features of MTR

The robots used were either exclusively telepresence robots (Giraff, VGO communication, Double, Kubi) or social assistive robots (Kompai, Scitos G3, Pearl), including a telepresence function as part of the services they offer (cognitive stimulation, appointments and medication reminders, games).

Most of the robots were in phases where their development and implementation were still ongoing. The MTR usability was tested in most of the studies. On the one hand, the robots whose only functionality was telepresence were, for the most part, guided by an external intervener (family or professional) and had simple interface, which was easy to use for OAs [19,26,34,35,36,40,42,44,45,46,47,48,51,53]. In the publication by Niemela et al. [48], for example, the use of the robot was appreciated by OAs because the only task required was to press a button on the robot to accept and initiate a conversation with their families. On the other hand, robots that had multiple functionalities required manipulation through complex interfaces that OAs with cognitive impairments had difficulty achieving [19,37,38,39,43,49,50,52,54,55]. In two publications, these interfaces were evaluated and modified after user feedback to make them more intuitive and better adapted to the abilities of older people with cognitive impairments [38,43].

Several authors also experimented with the robot’s movement and positioning in relation to OAs in order to promote satisfactory interaction [19,36,40,43,45,46,47,51]. Navigation could nonetheless additionally be difficult [45] or time-consuming [42] for family carers.

Some users (elderly people, family carers, and professionals) expressed concerns about the big size of the robot, with this being a potential problem in some dwellings [26,42]; its noise level [42]; and power consumption [42]. Bugs in the robot’s functioning, such as disconnecting, could cause discomfort to users [47]. Maintenance was also a source of concern, particularly with regards to battery durability [26,42]. Use of the robot furthermore required an internet connection for patients and careers [47,48].

##### Training Needed to Use MTR

Authors insisted on the need to train participants in navigating the interface and moving the robot. Two authors insisted on the need to train participants (especially professionals and family members) in using the telepresence robot and navigating the interface [45,51]. Professionals and families expressed difficulty in handling the telepresence robot [19,51], particularly while installing it in front of the OA while interacting with him/her at the same time [36]. OAs who did not have to handle the telepresence robot had a simple training, for instance, learning to press a button to accept the call [48].

#### 3.2.3. Safety (SAF)

##### Patient Safety with MTR

In terms of safety, most of the authors insisted on the robots’ abilities to avoid obstacles. In the work of Cesta et al. [40], some participants expressed a fear of being jostled by the robot and of falling, especially during the first contact with the robot. However, this fear no longer seemed to be present when OAs interacted with robot [26] for more than a year.

Interactions with robots seemed to have few adverse side effects. Several authors noted that OAs did not report anxiety while interacting with robots [39,54]. Cesta et al. [26] showed that interactions with robots caused anxiety (observable on anxiety rating scales) and an increase in hearth rate in healthy OAs as well as those suffering from mild cognitive impairment. This was identical between the two groups and was not considered to be alarming. For these authors, this increase in heart rate was more a reflection of an increased state of alertness related to the interaction with the robot than a negative side effect. Moyle et al. [47] reported that some patients suffering from dementia or institutionalized patients may be frightened or confused by the robot.

#### 3.2.4. Clinical Effectiveness of Robotic Interventions (EFF)

##### Patient Satisfaction with MTR

In most of the publications, authors noted that participants were satisfied with the robot’s appearance. Manufacturers often favor a physical form comprising of anthropomorphic and mechanical characteristics. Additionally, robots were not considered intimidating for people as long as their size did not exceed 1.5 m [39].

Most authors noted high feasibility of experimentation with MTR. Healthy OAs found them practical [26] and interesting, and enjoyed interacting with them [26,36,38,50]. This group of users showed a high level of commitment to the proposed activities, whether they be telepresence functionalities [26] or the other functionalities of the robot (appointment reminder, cognitive stimulation, etc.) [36,39,45,49,50]. Interactions with the robot were also appreciated by OAs presenting cognitive disorders, be it MCI [53] or dementia [47]. Caregivers and professionals also expressed their satisfaction in interacting with OAs using an MTR as a support [36,49,50,51]. The playfulness of the interaction with the robot was mentioned by several authors [50,54].

##### Health-Related Quality of Life and MTR Use

Some authors assessed the effect of MTR on OAs’ loneliness and isolation and reported a reduction of these two dimensions either with a qualitative assessment using interviews [19,26,35,45,47,48,51] or with scales and questionnaires [26,35,45]. In addition, in many papers examined, the authors mentioned that MTR use was able to increase communication, interaction, and connectedness between OAs and family members or professionals (as detailed in the following paragraphs).

According to some OAs, the MTR was useful [42]. It facilitated communication with their families [42] and was considered more interactive and attractive than traditional telephone calls [19,36,42,45,47]. It reduced feelings of loneliness by reducing perceived distance with their families [19,45], increased feelings of security [26,36,41], and improved their day-to-day mood and quality of life [41].

The robot gave caregivers a sense of closeness to OAs [42]. It gave them the feeling that the OAs were not isolated in their home [16,47] or in nursing homes [47]. It enabled them to reduce the number of travels [47] dedicated to face-to-face visits to OAs. The use of the robot fostered the development of family ties with professionals in the institution [47,48]. Finally, the robot enabled them to reduce feelings of guilt about not going to see the OAs as often as necessary [42]. Caregivers also considered that the MTR could contribute to improve OAs’ health, wellbeing, cognitive functioning, and control of everyday activities such as medications and safety, and thus reduce families’ mental and physical workload [41].

Longer-term benefits were also observed in a small number of patients. In the study by Niemela et al. [48], family members and professionals noted improvement in the OA’s well-being following regular interaction with relatives via the robot during a 12-month period. The OA, however, preferred to use the telephone to communicate with her relatives due to hearing problems. In a study by Cesta et al. [26], during which a robot was installed in the home of an elderly couple for one year, the authors found that the OAs and their son, who was their caregiver, were satisfied with the presence of the robot, found it useful for communication, and did not get tired of it over time.

The benefit of MTR on communication has also been evaluated in two randomized studies. Broadbent et al. [37] compared a group of OAs and professionals who encountered and interacted with robots placed in common areas for 12 weeks to a control group. They showed no benefits on different scales of depression, mobility, and quality of life. In a randomized study, another randomized study was carried by Moyle et al. [47], who studied the benefits of interactions between five OAs suffering major neurocognitive disorders in institutions and their families for 6 weeks. The authors observed that the OAs expressed positive emotions and had a high level of engagement while interacting with their loved ones, while the families found that interaction with the robot reduced social isolation.

##### Morbidity and MTR Use

The robot was also useful in healthcare settings when assisting in diagnosis and follow-up [19,34,36]. For instance, in two works [34,36], frail OAs with chronic diseases were followed-up by professionals with virtual visits and interviews using the MTR in order to check their health and detect any worsening of their medical condition.

In a randomized study, Bakas et al. [34] studied the benefits of interactions with a telepresence robot that was manipulated by student nurses from a distance for 3 weeks for 21 people in a care home (11 in the group with the robot and 10 in the group without), noting an improvement in number of bad days, depressive symptoms, sleep, quality of life, and older people’s confidence in managing their health.

#### 3.2.5. Organizational Aspects (ORG)

##### Health Delivery Process Using MTR

The technology could affect the work processes in the health and care institutions because it is time-consuming [34,36,45]. In practice, the implementation of day-to-day usage and interface navigation proved to be a major constraint for professionals and family members because of their already very busy schedules [34,36]. Therefore, family members often preferred to use the telephone instead of the robot to contact their relatives [48]. Teaching, training, and use of MTR by a large number of users with varying educational backgrounds and computer skills were also time-consuming and difficult to implement [36].

##### Culture (Professionals MTR Acceptance)

Professionals were satisfied that the OAs in nursing homes enjoyed interacting with the robot [42,47]. They felt that the robot could assist them [49] and be a useful tool for diagnosing and monitoring OAs’ health [19,26,42,45]. Professionals were nevertheless concerned about the difficulties linked to potential emergency interventions [36]. They furthermore were reluctant to consider robots as a companion for OAs [26].

#### 3.2.6. Medico-Economic Aspects (ECO)

##### Resource Utilization

Financial aspects were mentioned in two papers [19,47]. Participants in the study by Beer et al. [19] wanted to know the cost of the robot and whether acquiring it for the home would be financially feasible. Moyle et al. [47] discussed the need of considering the cost-effectiveness of MTR and examining their advantages regarding other non-robotic communication technologies (such as tablet computer or web conference applications). These authors also stressed the importance of conducting a cost analysis in future studies examining the human and physical costs related to the use of a robot, both for its purchase and its maintenance.

#### 3.2.7. Ethical Aspects (ETH)

##### Benefit–Harm Balance

The benefit–harm balance was discussed by some authors [19,26,47,48,54]. On the one hand, the participants considered that the MRT use fostered communication and interactions between OAs and families. On the other hand, they feared the lack of real physical contact related to the decrease in family visits following the installation of the robot [19,42], and the fear that robots would replace humans.

##### Autonomy

The OAs in the study by Cesta et al. [26] wished to keep control over the robot’s actions. They asked for an option to refuse or end a call as they wished, and an option to control who calls them [19,26]. Furthermore, some OAs suggested the implementation of social rules for proper and polite use of the system to make sure that OAs were in control of their interactions with other persons [19].

##### Respect for Persons

Some participants (caregivers, professionals) expressed concern about loss of privacy, especially in health and care institutions [19,36,40,47]. Niemela et al. [48] suggested the implementation of a recommendations guide with regards to the use of MTR. For instance, family members would be welcome to use the MTR in the room of the OA they care for but would not be allowed to use the robot in common spaces such as the dining room where other OAs might fall within the field of view of the robot without having requested it or accepted it.

#### 3.2.8. Patients and Social Aspects (SOC)

##### OAs’ Perspectives on MTR

The robot was highly appreciated by some healthy OAs who had never used a computer before [55]. The sessions with the robot were pleasant and enjoyable for the OAs and gave them a positive view of technology and an incentive to increase their knowledge in this area. After using the robot, they wished to create an e-mail account, buy a computer, or enroll in a computer course.

Regarding robot acceptance, three authors reported positive results in terms of perceived usefulness and intention to use the tool by OAs, family members, and professionals after a single test session [19,45,51]. However, participants in the study by Gertowska et al. [41], who reported good rates of robot acceptability during one session, stated that several weeks or months of interactions were necessary to judge its long-term acceptability. In two studies [26,42] that lasted at least 12 months, participants demonstrated good social and functional acceptance of the robot and expressed a wish to continue the experiment beyond the initially proposed period.

However, in a study of 11 elderly participants with no or mild cognitive impairments, Wu et al. [54] showed low levels of intention to use a social telepresence robot that was perceived to be of little use in daily life and linked to the projection of a stigmatized image of themselves [54].

Baisch et al. [35] compared the interactions of 29 healthy OAs with robots Giraff (Giraff Technologies AB: Västerås, Sweden) and Paro (National Institute of Advanced Industrial Science and Technology: Tokyo, Japan). For Giraff, acceptance was satisfactory for people who had little social support but were able to control the robot themselves. Acceptance was low for people who were not able to control the robot. For these elderly people, Giraff had a less stigmatizing image than the Paro, a baby seal robot.

##### Communication Aspects on MTR

Several authors insisted on the need to prepare participants by describing the robot’s capabilities and encouraging them to have realistic expectations of it [38]. According to Stafford et al. [52], robot acceptance and intention to use were inversely correlated to high expectations, people expecting robots to be able to perform many actions on their own, and being ultimately disappointed by reality and usage limitations.

The benefit of familiarizing users with the robots (video and face-to-face presentation of the robot) before the test session was indeed noted by three authors [38,39,52]. In particular, OAs wanted information on the following points before using the robot: cost of the robot, system capacity and functionality, confidentiality aspects, operating mode and operating manual, maintenance methods, safety, and system limitations. Two authors noted that OAs had an increasingly positive attitude towards the robot as they became familiar with it and used it repeatedly [50,52].

#### 3.2.9. Legal Aspects (LEG)

Aspects related to rules and regulations were not described in the publications reviewed.

Table 4 shows a summary of the benefits and barriers to the implementation of robots in daily practice.

## 4. Discussion

MTR is a developing field, with a variety of uses and emerging features. These robots have potential for maintaining social ties and reducing loneliness in OAs. Indeed, these tools could help OAs experience the presence of a readily available social network and reassure them. Moreover, a robot, which is a physical entity in its own right, could provide OAs with a sense of security and presence.

This review identified and summarized the methodology and findings of 24 studies in which the authors examined the use of telepresence robots in OAs and their families and professional careers. This analysis of the literature showed that OAs enjoyed interacting with these tools (high engagement in the activity) with few side effects (low anxiety level) and recognized their potential societal impact. Nonetheless, a number of technical, efficiency, organizational, economic, ethical, and sociological barriers remain to be overcome before these tools are widely disseminated and used by a large number of older people on a daily basis.

Robot usability could firstly prove to be a barrier. Indeed, several authors showed that healthy OAs and those suffering from cognitive impairment (mild cognitive impairment and mild dementia) could enjoy using robots provided they were easy to use. This ease of use was essential for OAs. User experience should also be simple for professionals and families. The videoconferencing modality in MTR was generally a simple and powerful feature and provided significant user satisfaction. System navigation could nonetheless prove to be complex for family members and professionals, requiring implementation of training and being a source of device abandonment [34,36,45].

As presented in the results, some authors reported a reduction of OAs’ loneliness and isolation with MTR. In addition, other authors showed the benefit of MTR for communication and interactions between OAs, family caregivers, and professionals, which might imply an indirect benefit on loneliness and isolation. However, there is still a lack of evidence regarding the effectiveness of MTR in alleviating the problems in OAs. Other authors have examined the impact of MTR in OAs on other dimensions such as stress and depression [34], quality of life [37], confidence [35], and feeling of security [36,47]. In a review of the literature, Shishegar et al. [25] rated the benefits of various social robots and concluded that telepresence robots were the most beneficial group of robots after companion ones. Some of the results are promising. There is nevertheless a need for further studies in healthy OAs and those with cognitive impairment to confirm the value of these tools in the short and long term and to investigate which populations could benefit most.

Organizational aspects can be a barrier to the implementation of such robots. Several authors have highlighted two essential but time-consuming steps for implementation: firstly, training OAs to use MTR through presentations and demonstrations, and secondly, training professionals and families in the use of these tools [34,47]. Moreover, various aspects relating to the implementation of these interventions in everyday life have yet to be specified. These include guideline to familiarize and train users to use of these robots.

Little work has been devoted to the medico-economic aspects regarding this kind of technology, which constitutes a critical dimension for implementation (costs, healthcare funding, cost-effectiveness, willingness to pay, reimbursement, etc.). Indeed, the authors noted that many potential users were wondering about the cost of the robots and the economic model to adopt concerning their use.

Several ethical questions further arose and necessitate the implementation of good care practices for the use of robots. It is namely essential to have the consent of the OAs before using the robot. Several authors noted that OAs requested to control the robot themselves and wanted to be able to decide when they initiated the interaction with their relatives and when they stopped it. Moreover, rules regarding confidentiality need to be specified in the home and in institutions. Indeed, in the study by Niemela et al. [48], participants insisted on the fact that it was not appropriate for relatives to be able to move the robot in common areas and thus see OAs other than their relatives, who had not given their permission to be seen. In addition, the question of the presence of relatives via the robot during care procedures by professionals was raised. Some authors have also emphasized the risk of replacing real family visits with virtual ones. One ideal mode of mitigating this risk might be to combine the two modes of interaction. A satisfactory way to reduce this risk would be to alternate face-to-face and virtual modes of interaction, for example, real and virtual visits by families, or face-to-face, and remote consultations by professionals.

It also seems important to raise the issue of equal access to the use of these robots in order to define the target populations, for example, the most isolated and the most deprived people, who could best benefit from these psychosocial interventions. It will be necessary to examine these aspects during the implementation of public programs intended to support the elderly through this type of technology and to diversify the offers of psychosocial interventions so that they can be both face-to-face and remotely.

Recommendations remain to be identified for the protection and safety of users. Indeed, most of these tools are connected to the Internet and can give access, if not protected, to personal information that may have been recorded by cameras and microphones. Before any use, it is important that the new digital tools are secure, that their deployment in homes is thought out, and that certain recommendations for use are defined. In order to progress in this field, Devillers (2017) [56] proposed a set of rules of ethical and safe behaviors for robots, such as “You will not disclose my data to anyone”, “You will forget everything I ask you to forget”, and “You may disconnect from the Internet if I ask you to do so”.

The question of acceptance is crucial in the literature. Findings from this review of the literature may allow us to conclude that OAs, family members, and professionals showed good acceptance of MTR. However, their acceptance of MTR in the long term remains to be explored.

One of the robot acceptance models proposed by Young et al. [57] proposes three phases in the acceptance of robots: the first includes contact with the robot, manipulation (user experience), and integration of the robot in a person’s particular context and environment. The first contact with the robot is essential when it comes to a user’s acceptance of it [57]. Indeed, the robot if unpleasant to the user may be immediately rejected. For example, a robot that looks too humanlike may cause a feeling of discomfort [58], while a robot that looks too mechanical is unlikely to be considered a possible companion [54]. On the contrary, a pleasing appearance will increase a person’s commitment to the robot. Our results show that the robots developed by manufacturers in recent years with an appearance which lies between that of a human and machine were satisfactory for users.

The second step described by Young et al. [57], pertaining to the handling of the robot and its functionalities, is also acceptable to users. The analysis of the literature showed that the OAs enjoyed watching and interacting with the robot during sessions, although they expressed the need for more user-friendly interfaces, less technical problems, or integration of advanced functions such as psycho-emotional capabilities to naturally interact with the users [55]. The finding that some OAs who had never used computers before the MTR sessions were willing to use communication technology after manipulating the robot is in line with previous work showing that OAs’ attitudes toward a technology (computer, robot) could be improved over time though direct experience with it [54,55]. Thus, we may suggest that robot sessions, which are pleasant and funny experiences for OAs, could contribute to make them more confident to use technology and be a promising approach to reduce generational digital divide.

The third step, however, can be a source of hindrance when it comes to the acceptance of the robot. Indeed, although people enjoy interacting with the robot during one or more sessions, especially in the laboratory, their acceptance in the longer term and at home remains to be clarified. Indeed, when people were asked about their willingness to use the robot in the long term, the results were divergent according to the publications. In some studies [19,26], healthy OAs stated that they were willing to use the robot in the long term, while in others [54], OAs did not wish to have a robot at home. The first reason given was that the robot did not seem to be useful for them as they were not physically or cognitively dependent. They nonetheless reported that robots seemed useful either for dependent people or in the future when they themselves would become dependent. The second reason given was that social robots risked reducing social contact by replacing genuine human contact with virtual contact via videoconferencing. The risk would then be of dehumanization, with the possibility of feelings of abandonment, particularly felt by isolated people. The third reason given was that these robots could be stigmatizing in that they designated their users as potentially dependent [54].

The COVID-19 crisis and its consequences on physical distance and “stay at home” recommendations could potentially facilitate users’ acceptance of TMR. Indeed, the constraints linked to face-to-face encounters have made it necessary and relevant to develop technical solutions, social robots in particular, to overcome these difficulties. In several reviews [16], the authors have noted that these tools could make it possible to maintain contact with family and friends in order to share life moments despite the distance, and to limit unnecessary travel while reducing social isolation.

One example features a telepresence robot that was used to greet patients presenting for care in a clinic in Brazil. Healthcare professionals observed that patients were particularly attentive to the robot’s instructions, and followed its advice, suggesting that patients were engaged in the interaction with the robot, had a good level of acceptance, and considered it reliable [59]. In France, in the Auvergne-Rhône-Alpes region, telepresence robots, intended for the region’s high schools, have been mobilized and implemented in care homes. This system has enabled not only families to visit their relatives virtually, but has also facilitated exchanges with music therapists, religious representatives, and volunteers [60]. In Italy, a telepresence robot has facilitated the monitoring of medical equipment in patients’ rooms. Initially, patients, especially the elderly, did not appreciate the interactions with the robot, but then came to appreciate the benefits (e.g., increased exchanges with doctors) that these tools could bring to their care [61]. Thus, the COVID-19 virus could increase users’ perceived sense of usefulness of telepresence social robots and, consequently, the acceptance of their use. It would therefore be necessary to assess in further studies whether the pandemic may have changed older people’s perception of assistive and telepresence robots by making them more useful and less stigmatizing. One possible work might be to assess MTR acceptance in OAs, caregivers, and professionals at home for a long period during the COVID-19 pandemic. It would be interesting to examine whether the use of a robot gives OAs an incentive to use other communication technologies such as computers or tablets and thus has a beneficial impact on generational digital divide.

### Limitations of the Study

This review makes an interesting contribution to the state of the art work on telepresence robots for the elderly. We did a systematic review of the literature and included all studies that included experimentation regardless of protocol in order to allow an exhaustive synthesis of the point of view of these individuals, families, and professionals on the benefits and current barriers to the use of these tools. This review has several limitations. We have included only English language publications. We were able to omit posts in which people manipulated a social robot whose communication and social interaction functionality did not appear as a keyword or in the summary. Most of the studies included no comparators or baseline assessment, being mostly short-term studies with limited sample sizes. Heterogeneity of telepresence robot appearance and functionalities prevent general conclusions.

## 5. Conclusions

The isolation of older people is a growing phenomenon that requires appropriate care. Currently, the development of innovative technologies promoting the maintenance networks could help to combat the process of social withdrawal. In this respect, the deployment of social telepresence robots in the homes of the elderly or in institutions could be a major asset. Overall, we have found that these new tools are accepted, that they are considered to be usable, that their appearance suits users, and that they facilitate contact. However, some barriers to their being implemented remain, such as occasionally overly complex interfaces, need for time for learning and training, and risks of dehumanization.

In the literature, we have not been able to find an overall trend in the results, which remain heterogeneous due to the lack of similarity in the studies. Further research is therefore necessary, both to measure the impact of telepresence robots on the social isolation of elderly people in the long term but also to define more precise implementation recommendations pertaining to various technical, efficiency related, organizational, economic, ethical, and sociological aspects.

## Figures and Tables

**Figure 1 ijerph-18-03597-f001:**
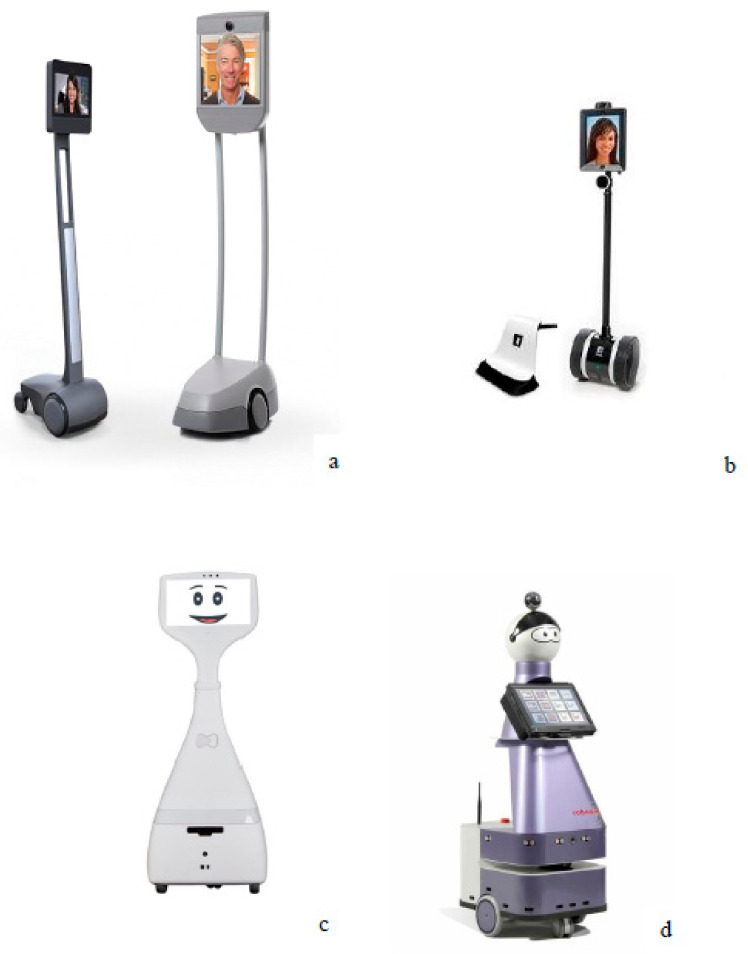
Examples of telepresence robots: (**a**) Beam + [21]; (**b**) Double 3 [22]; (**c**) Cutii [23]; (**d**) Kompai [24].

**Figure 2 ijerph-18-03597-f002:**
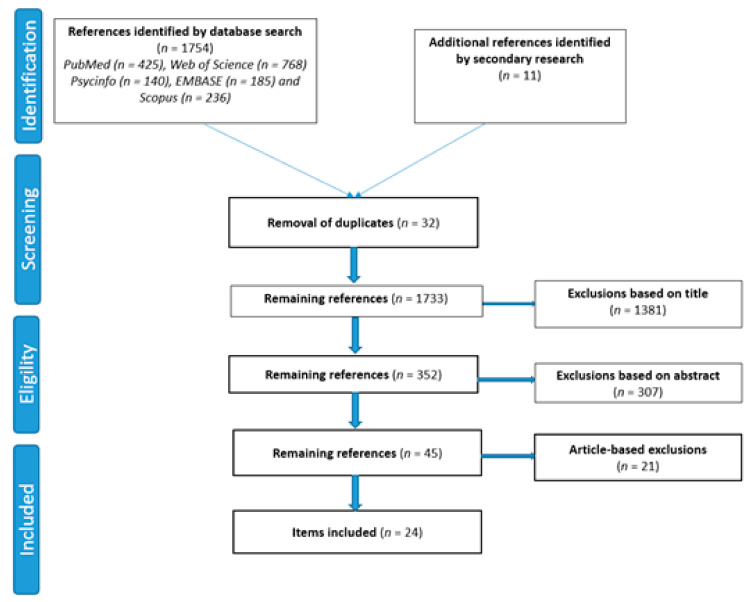
Flow diagram.

**Table 1 ijerph-18-03597-t001:** Domains of assessment of the health technology assessment (HTA) core model version 3.0 (EUnetHTA Joint Action 2, 2016) [32].

Domains	Main Features
Health and current use of the technology (CUR)	A description of the condition targeted by the technology, the therapeutic purpose of the intervention, and the current standard treatment to address it.
Description and technical characteristics of technology (TEC)	A description of the technical features of the technology, its level of maturity, the resources (material, infrastructural, etc.), and skills required to use it.
Safety (SAF)	A description of the risk and unwanted effects caused by the technology, and the way to prevent and manage it.
Clinical effectiveness (EFF)	A description of the effects of the intervention on the ability to reach the clinical objectives set for the intervention, on the condition of the quality of life and the autonomy of the users, as well as on the follow up conduct by the professionals who take part in the intervention
Costs and economic evaluation (ECO)	A description of the costs, the health-related outcomes, and economic efficiency of the technology.
Ethical analysis (ETH)	A description of issues related to ethic and values when using the health technology.
Organizational aspects (ORG)	A description of the allocation of resources (material artefacts, skills, knowledge, money, work culture, etc.) required to implement the technology in the organization and the healthcare system.
Patients and social aspects (SOC)	A description of the representations conveyed by the intervention at the individual’s and collective’s levels, for the patients, their entourage, the caregivers, and society as a whole.
Legal aspects (LEG)	A description of regulations and laws to be considered in evaluating a technological intervention.

**Table 2 ijerph-18-03597-t002:** Studies description.

Study	Country	MTR Model (Manufacturer)	Setting	Time Period	Assessment Objective
Bakas et al. (2018) [34]	USA	VGO Communications (VGO)	Home	3 weeks	Study 1: Feasibility Study 2: Clinical impact in a randomized controlled trial (2 groups, with and without robot)
Baisch et al. (2017) [35]	Germany	Giraff (GiraffPlus) and Paro (national institute of advanced industrial science and technology)	Laboratory	1 day	Technology acceptance (investigation of the influence of psychosocial factors)
Beer et al. (2011) [19]	USA	MTR Texai project (Willow Garage)	Laboratory	1 day	Technology acceptance and usability
Boman and Bartfai (2014) [36]	Sweden	Giraff (GiraffPlus)	Hospital	1 day	Usability and user experience
Broadbent et al. (2016) [37]	New Zealand	Guide and Cafero	Senior housing	12 weeks	Clinical impact in a controlled trial (2 groups, with and without robot); technology acceptance; organizational impact
Caleb-Solly et al. (2018) [38]	England and the Netherlands	Kompai (Kompai Robotics)	Laboratory	2 days	Usability and user experience
Cavallo et al. (2018) [39]	Italy	Robot ERA-Scitos G5 (MetraLabs)	Laboratory	1 day	Technology acceptance
Cesta et al. (2012) [40]	Italy	Giraff (GiraffPlus)	Laboratory	1 day	Technology acceptance and usability
Cesta et al. (2016) [26]	Italy	Giraff (GiraffPlus)	Laboratory	12 months	Clinical impact; technology acceptance and user experience
Gertowska et al. (2013) [41]	Poland	Robot assistant for MCI patient at home (RAMCIP)	Hospital	1 day	Technology acceptance and usability; social impact
Gonzalez-Jimelez et al. (2013) [42]	Spain	Giraff (GiraffPlus)	Home	12–18 months	Technology acceptance
Granata et al. (2013) [43]	France	Kompai (Kompai Robotics)	Laboratory	1 day	Usability and user experience
Hiyama et al. (2017) [44]	Japan	Double (Double robotics) and Kubi (Xandex In)	Laboratory	5 days	Technology acceptance and usability
Koceski and Koceska (2016) [45]	Macedonia	MTR (academic research)	Nursing home	1 day	Technology acceptance
Kristoffersson et al. (2014) [46]	Sweden	Giraff (GiraffPlus)	Laboratory	1 day	Usability (positioning of the robot)
Moyle et al. (2014) [47]	Australia	Giraff (GiraffPlus)	Long-term care unit	4 months	Feasibility and technology acceptance
Niemela et al. (2019) [48]	Finland	Double (Double robotics)	Nursing home	12 weeks	Technology acceptance and user experience
Pineau et al. (2003) [49]	USA	Nursebot Pearl (academic research)	Nursing home	1 day	Feasibility and technology acceptance
Schroeter et al. (2013) [50]	The Netherlands	Scitos G3 (MetraLabs)	Laboratory	2 days	Technology acceptance and usability; social impact
Seelye et al. (2012) [51]	USA	MTR-VGO system (VGO)	Home	2 days	Technology acceptance and usability
Stafford et al. (2014) [52]	New Zealand	Healthbot (Yujin Robot)	Senior housing	2 weeks	Feasibility and technology acceptance
Tiberio et al. (2012) [53]	Italy	Giraff (GiraffPlus)	Laboratory	4 days	Clinical impact (psychophysiological responses to the robot)
Wu et al. (2014) [54]	France	Kompai (Kompai Robotics)	Laboratory	4 weeks	Technology acceptance
Zsiga et al. (2017) [55]	Hungary	Kompai (Kompai Robotics)	Home	2–4 months	Technology acceptance and usability

MTR = mobile telepresence robots; MCI = mild cognitive impairment; RAMCIP = Robot assistant for MCI patient at home.

**Table 3 ijerph-18-03597-t003:** Selected pertinent articles to the subject matter.

Study	Population			Assessment Indicators (Method)	Benefits of MTR Ise	Impact on Social Isolation	If Yes, Which	Barriers to MTR Use
	Older Adults (OAs)	Professionals	Family Members					
Bakas et al. (2018) [34]	Polypathological OAs Study 1, *n* = 5 Study 2, *n* = 22	Nurses (*n* = NP)	NA	Number of “bad days”, depression, stress, fatigue, pain, shortness of breath, sleep, quality of life, confidence pre- and post-intervention (scales)	Good feasibility; improvement in the number of “bad days”, depression, sleep, quality of life, confidence in managing one’s own health	No	NA	Training of nurses to handle the robot’s displacement
Baisch et al. (2017) [35]	Healthy OAs (*n* = 29)	NA	NA	Loneliness, depressed mood, life satisfaction, social support (scales)	Regarding Giraff, good acceptability for AOs with limited social support who can control the robot; regarding Paro, no association between acceptability and psychosocial variables.	Yes	Improvement of social contact but reduction of emotional impact compared to personal visit	Regarding Giraff: lack of autonomy (is easily rendered useless if the help of a third party is needed to handle it); it is difficult for the main user to have full control of the robot
Beer et al. (2011) [19]	Healthy OAs (*n* = 12)	NA	NA	Perceived benefits and concerns; suggestions for use cases, recommendations on system design (semi-structured interviews)	Positive feedback from the camera device, helps reduce travel, can provide assistance in health diagnostics; expressed desire to use the robot in the future.	Yes	Reduction of social isolation	OAs concerns: lack of privacy, lack of real contact, ease of use, excessive or inappropriate use; expressed desire to know the capabilities and cost of the device before use
Boman and Bartfai (2014) [36]	OAs cognitive impairment (*n* = 3)	Nurses (*n* = 38); assistant night nurses (*n* = 10); occupational therapists (*n* = 3)	NA	Expectations, usability, and usefulness of the MTR (questionnaire, Likert scale, and open-ended interview)	OAs: very satisfied, easy to use and pleasant system, increases the feeling of security; Pro: positive experience	No	NA	Professionals: a lot of time for training—difficulties in handling the robot and interacting with the OAs at the same time, difficulties in emergency response, privacy concerns
Broadbent et al. (2016) [37]	OAs: healthy and with cognitive impairment (*n* = 52)	Care workers (*n*= 53)	NA	OAs: Depression, quality of life, mobility, activities of daily living (scales); Pro: job satisfaction, demoralization, attitude towards robots (scales)	OAs: no difference in the scale scores between the two groups; positive, neutral, or negative reactions and opinions of robots; Pro: rather positive opinion of robots	No	NA	Robots difficult to use in OAs with cognitive deficit or motor disability
Caleb-Solly et al. (2018) [38]	OAs: healthy and with cognitive impairment (*n* = 11)	NA	NA	Usability (questionnaire), satisfaction, perceived usefulness, privacy concerns (semi-structured interviews)	Adequate usability and acceptance	No	NA	Need to prepare users for the real capabilities of the robot: many technical constraints, need for realistic expectations towards robot use.
Cavallo et al. (2018) [39]	Healthy OAs (*n* = 45)	NA	NA	Acceptance, perceived robustness (semi-structured interviews), questionnaire of appearance	Good acceptance of robots; appearance and services appreciated, no privacy concerns, no anxiety about using the robot	No	NA	Previous familiarization necessary, importance of combining anthropomorphic and machine features for robots, appropriate robot size (150 cm)
Cesta et al. (2012) [40]	Healthy OAs (*n* = 10)	Nurses (*n* = 26)	NA	Technology acceptance, usability, satisfaction, positive and negative aspects (focus groups, interviews)	Good engagement with the robot, pleasant to see, satisfactory navigation, gives a feeling of security, interaction with it is spontaneous	No	NA	Concern about size and battery, confidentiality, ability of MTR to avoid obstacles and return to its charging station
Cesta et al. (2016) [26]	OAs with health concerns (*n* = 2, a couple)	NA	Adult child (*n* = 1)	OAs: loneliness, social support, service satisfaction, depression, emotions, usability, acceptability, psycho-social impact, telepresence dimension, user expectations and attitude towards the robot (scales and questionnaire); family: affects, usability, telepresence dimension, psychosocial impact, expectations and attitude towards the robot (scales and questionnaire)	Good social and functional acceptance by OAs and family; no loss of interest over time; wish to continue the use of the robot beyond 12 months	Yes	MTR appreciated for its ability to create company and alleviate loneliness	Concern about MTR management and maintenance, wish expressed to have more control over the robot
Gertowska et al. (2013) [41]	Healthy OAs (*n* = 10), OAs with MCI (*n* = 8)	NA	NA	Usability, acceptability, and societal impact (questionnaires)	Satisfactory acceptability and perceived social impact; helps reduce the burden on caregivers; improves the patient’s daily life by facilitating communication; improving safety, mood, and quality of life	No	NA	Necessity of a long-term interaction to evaluate the subjective value of the robot
Gonzalez-Jimelez et al. (2013) [42]	OAs (*n* = 3)	Professional team of a health center (*n* = NP)	Some relatives (*n* = NP)	Usability, acceptance, and user experience (interviews and questionnaires)	OAs: good usability and acceptance of the robot; families: feeling of being closer to the OAs; Pro: benefit of being able to follow the health status of patients	No	NA	Concerns about usability, risk of losing “real” contact with OAs; concerns about the size, power consumption, and noise of MTR
Granata et al. (2013) [43]	Healthy OAs (*n* = 11), OAs MCI (*n* = 11)	NA	NA	Usability (questionnaire and observations)	Better performance for healthy, younger, and IT-experienced OAs	No	NA	NA
Hiyama et al. (2017). [44]	Healthy OAs (*n* = 15)	Lecturers and assistants of a lifelong learning service (*n* = NP)	NA	Acceptance and usability (questionnaire and observations)	Good acceptance of the robot, easy communication between teachers and OA class during distant class learning	No	NA	NA
Koceski and Koceska (2016) [45]	OAs with no severe disability (*n* = 30)	Professional caregivers (*n* = 5)	NA	Perceived usefulness and ease of use (questionnaire)	Good acceptability of the basic robot functionalities, willingness to use the robot in the social and medical fields	Yes	The robot helps to reduce loneliness by bridging distances and facilitating communication	Requires training to learn how to manage MTR navigation
Kristoffersson et al. (2014) [46]	Healthy OAs (*n* = 10)	NA	NA	Robot positioning experience with respect to the OAs (Interview and obsevations)	When using MTR, it is important for OAs to have eye-contact with the person embodied, training on the positioning of the robot for pilot users is important	No	NA	NA
Moyle et al. (2014) [47]	OAs with dementia (*n* = 5)	Care workers from a long-term care facility (*n*= 7)	Family caregivers (*n* = 6)	Feasibility, emotional state, and engagement while using the robot (semi-structured interviews, observational data)	OAs: Enjoyment and positive emotions when using MTR with a high level of engagement; family and Pro: positive experience, increased contact with family, helps to feel reassured	Yes	Helps to reduce social isolation and increase connection between residents and families, especially for participants who lived some distance away or do not see each other regularly	Technical problems: robot errors, internet connection; ethical issues: confidentiality; need to make a cost analysis.
Niemela et al. (2019) [48]	OAs with pathology (*n* = 1)	Nurses (*n* = 3)	Adult children (*n* = 2)	User experience, technology acceptance (pre/post-experimentation interviews, user observations, user journals)	OAs: enjoyment of the family presence; family: satisfaction of seeing the OAs with respect to the only voice calls; Pro: satisfaction when seeing the patient’s enjoyment	Yes	Reduction of social isolation, increased connection between OAs and family	Risk of OA confusion, lack of real physical contact, lack of control over the device, privacy concerns
Pineau et al. (2003) [49]	OAs with MCI and other limitations (*n* = 6)	NA	NA	Feasibility and technology acceptance (observations and post-experimental interviews)	Predominantly positive feedback from OAs, positive conclusion of the robot’s role in assisting nurses	No	NA	Need for technology that adapts to individual differences
Schroeter et al. (2013) [50]	OAs with dementia or MCI (*n* = 6)	NA	Family caregivers (*n* = 5)	User experience, technology acceptance (post-experimentation interviews, observations, user journals)	Good usability, acceptability, and social impact	No	NA	NA
Seelye et al. (2012) [51]	Healthy OAs (*n* = 8) and MCI OAs (*n* = 1)	NA	Relatives (*n* = 8)	Technology acceptance, user experience, usability (interviews)	OAs: positive experience; appreciation of the potential of robots to improve physical health, well-being, social connectedness, and autonomy; family: ease of installation and setup, mobility of the robot appreciated, increased feeling of reassurance	Yes	Good potential to increase OAs’ social connectedness	Operation of the handheld remote confusing for OAs, robot’s wheels not always adapted to handle transitions between different types of flooring; robot not usable by OAs with MCI
Stafford et al. (2014) [52]	OAs (*n* = 25)	NA	NA	Feasibility of robot deployment, feedback on the prototype and services, usability, psychological factors associated with the acceptance of robots (questionnaire)	Feasibility of deploying robots in OAs institutions; OAs having more positive attitudes towards robots, and those that perceived less agency in robot minds were more likely to use them	No	NA	NA
Tiberio et al. (2012) [53]	Healthy OAs (*n* = 9) and MCI OAs (*n* = 8)	NA	NA	Tolerance towards the robot and effects of the interaction with it (psychophysiological measures, scales, interviews)	Presence of the robot well accepted by healthy and MCI OAs: pleasant experience; good interest, level of attention, and participation	No	NA	Concern about MTR size (too big), real visits preferred to virtual visits
Wu et al. (2014) [54]	Healthy OAs (*n* = 5) and OAs MCI (*n* = 6)	NA	NA	Technology acceptance (questionnaire, semi-structured interview, focus group)	Robot found easy to use; non-threatening and fun	No	NA	Low intention to use the robot, perceived as not very useful for daily life use, negative image associated with MTR use
Zsiga et al. (2017) [55]	OAs with some mobility limitations (*n* = 8)	NA	NA	Technology acceptance, user behavior and experience (questionnaire, logs collected by the robot)	OAs considered mobility, entertainment, and obstacle detection to be the best robot functionalities.	No	NA	Low reliability of the robot, lack of 24/7 operation time; initial instability of speech recognition, navigation, and self-localization problems

Note: MCI = mild cognitive impairment, *n* = number, NP = not precised, NA = not applicable, OAs = older adults, Pro = professionals.

**Table 4 ijerph-18-03597-t004:** Summary of benefits and barriers to the implementation of robots in daily practice (HTA: health technology assessment).

HTA Dimension	Benefits of MTR	Barriers to MTR Implementation
Health problem and current use of the technology	Usable by all the OAs	Lack of recommendations according to health condition
Description of the technology	Pleasant design	Complex interfaces, technical problems, fear to fail to use robots
Safety	Few side effects (anxiety, confusion)	Limits of current technological capabilities
Clinical effectiveness	Satisfying user experience	Insufficient demonstration of real benefit
Cost and economic evaluation	Medico-economic evaluation to be developed	
Ethical analysis	Potential interest in facilitating contacts	Risk of dehumanization, stigmatization, disappointment
Organizational aspects	Potential time saving for families and professionals users	Time required for training and implementation when in use
Patients and social aspects	Good user acceptability during the experiments	Different user opinions on long-term use
Legal aspects	Legal framework to be developed	

## Data Availability

No new data were created or analyzed in this study. Data sharing is not applicable to this article.

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
