# Peer review of "Social Telepresence Robots: A Narrative Review of Experiments Involving Older Adults before and during the COVID-19 Pandemic"

_ijerph, 2021, doi:10.3390/ijerph18073597_

Round 1
Reviewer 1 Report
The manuscript describes a narrative review of published studies on older adults interacting with mobile telepresence robots. Review findings are structured along the 9 dimensions of the HTA Core Model. The presented work is a valuable contribution to the research field, however, revision of the manuscript is recommended before publication:
- The Authors emphasize that they adopted the "multidimensional analysis framework" as recommended by the European network for Health Technology Assessment (EUnetHTA). However, the detailed guidance of EUnetHTA on specific Topics and Issues was not followed and review findings are presented only at the domain level of the EUnetHTA core model. The provided reference (ref 22) does not include even these domain descriptions. The interpretation of domain concepts is occasionally divergent from the guidance on the use of the HTA core model, e.g., for organizational aspects, only the time of care aspect is discussed in Table 3. User experience and user acceptance are overlapping concepts not defined in the manuscript and the related findings are presented under multiple core model domains. The Authors are recommended to present their findings at the Topics / Issues level of the EUnetHTA Core Model. Furthermore, use of the HTA Core Model, including information on its version, must be disclosed in the final product(s) as indicated in the HTA Core Model® Licence.
- Wording regarding the number of studies supporting the narrative conclusions is sometimes misleading. The Authors state e.g. that "In most of the publications, authors noted that participants were satisfied with the robot’s appearance [25–27]." Considering that altogether 24 studies were included, a finding of 3 studies should not be considered as a finding of "most of the studies". Similarly, the authors describe their findings based on "several publications" or "several authors" but refer to very few studies, or even to a single study in many cases. E.g., "In several reviews [13], ...". The Authors are invited to rephrase and tone down their narrative conclusions when supported only by a minor fraction of the identified studies.
- Section 3.1, "Experimental times ranged from 1 day to 18 months.": suggest also reporting the median (2 days).
- Limitations: suggest reporting additional limitations: most of the studies included no comparators or baseline assessment; mostly short-term studies with limited sample sizes; heterogeneity of telepresence robot appearance and functionalities prevent general conclusions.
English grammar check is recommended - e.g. "key words"; "faisability"; "homme"; "complexe"; "some of results".
Author Response
Answers to Reviewer 1 :
The manuscript describes a narrative review of published studies on older adults interacting with mobile telepresence robots. Review findings are structured along the 9 dimensions of the HTA Core Model. The presented work is a valuable contribution to the research field, however, revision of the manuscript is recommended before publication:
- The Authors emphasize that they adopted the "multidimensional analysis framework" as recommended by the European network for Health Technology Assessment (EUnetHTA). However, the detailed guidance of EUnetHTA on specific Topics and Issues was not followed and review findings are presented only at the domain level of the EUnetHTA core model.
Answer: The way HTA framework was used for this analysis was better explained ((lines 222-234; lines 369-372; lines 418-436). We added Table 1 in which the different dimensions are described. A topic/issue level analysis was added when it was possible (3.2. Description of studies using HTA dimensions including topics and issues when available; beginning line 303).
.
The provided reference (ref 22) does not include even these domain descriptions.
Answer: The reference was changed by :
World Health Organization. (2011). Health technology assessment of medical devices. https://apps.who.int/iris/bitstream/handle/10665/44564/9789241501361-eng.pdf (see in the references)
The Authors emphasize that they adopted the "multidimensional analysis framework" as recommended by the European network for Health Technology Assessment (EUnetHTA). However, the detailed guidance of EUnetHTA on specific Topics and Issues was not followed and review findings are presented only at the domain level of the EUnetHTA core model. The provided reference (ref 22) does not include even these domain descriptions. The interpretation of domain concepts is occasionally divergent from the guidance on the use of the HTA core model, e.g., for organizational aspects, only the time of care aspect is discussed in Table 3. User experience and user acceptance are overlapping concepts not defined in the manuscript and the related findings are presented under multiple core model domains.
The Authors are recommended to present their findings at the Topics / Issues level of the EUnetHTA Core Model.
Answer: we have now presented our findings at the Topics / Issues level of the EUnetHTA as suggested by the referee (please see 3.2. Description of studies using HTA dimensions including topics and issues when available; beginning line 303).
Furthermore, use of the HTA Core Model, including information on its version, must be disclosed in the final product(s) as indicated in the HTA Core Model® Licence.
Answer: Information on version and the license was added (line 239)
- Wording regarding the number of studies supporting the narrative conclusions is sometimes misleading. The Authors state e.g. that "In most of the publications, authors noted that participants were satisfied with the robot’s appearance [25–27]." Considering that altogether 24 studies were included, a finding of 3 studies should not be considered as a finding of "most of the studies". Similarly, the authors describe their findings based on "several publications" or "several authors" but refer to very few studies, or even to a single study in many cases. E.g., "In several reviews [13], ...". The Authors are invited to rephrase and tone down their narrative conclusions when supported only by a minor fraction of the identified studies.
Answer: we have rephrased and toned down our narrative conclusions when supported only by a minor fraction of the identified studies and we have added the missing studies (please see in the results (please see 3.2. Description of studies using HTA dimensions including topics and issues when available; beginning line 303).
- Section 3.1, "Experimental times ranged from 1 day to 18 months.": suggest also reporting the median (2 days). ”
Answer: this has been rectified according to the referee’s suggestion (please see section 3.1; line 271).
- Limitations: suggest reporting additional limitations: most of the studies included no comparators or baseline assessment; mostly short-term studies with limited sample sizes; heterogeneity of telepresence robot appearance and functionalities prevent general conclusions.
Answer: these limitations have been added according to the referee’s suggestion. (please see lines 771-774)
English grammar check is recommended - e.g. "key words"; "faisability"; "homme"; "complexe"; "some of results".
Answer: these errors have been corrected in the text
Reviewer 2 Report
Overview and general recommendations:
The paper focus on social telepresence robots and their role when interacting with older adults, in relation to aspects of loneliness and social isolation. The authors choose a literature review, published in the last 20 year, thus including per and post COVID-19 time. The retrieved publications were analysed using a multidimensional framework, based on the HTA Core Model Dimensions.
Results of the paper are important and add knowledge to the current framing of assessment of social robots.
The main strength of the paper is the analysis conducted using an HTA approach and methodology. This is in my perspective, the novelty that the paper can bring to, not only the field of robotics but also to HTA field. However, the authors do not provide a justification why this method is in general important and even more important why it was selected to be used in the literature review, applied to social robots. Due to the technology under assessment, the reason for choosing this methodological framework is of great importance and the link to it is missing.
When reading the paper, I identify several issues that the paper should address, and it does not. Since this will imply more than a mere language check or small complements, I recommend that a major revision is necessary. However, I strongly encourage the authors to consider my suggestions, complement the paper and re-submit it.
Below, in more detail, I explain my perspective and concerns.
Major comments:
- Abstract: the reason why the review of literature was conducted is missing. The aim of the paper should be explicit for the reader, already in the abstract.
- Line 48 to 71: This paragraph refers exclusively to data from a questionnaire applied in France and no other country’s data is presented. Since the paper does not focus on a French perspective or is limited to an analysis in France, I was wondering why only such data was presented. Results from such questionnaires are important, but when presented in the paper, a short international perspective would be more interesting.
- In addition to the previous comment, a short indication of the total number of people over 60 exist in France would be useful so the reader can better frame the presented percentages.
- The paper focusses its analysis on older adults. An important factor related to this segment of the population is the “generational digital divide”, as mentioned in line 59. However, this is the only time that such important topic is approach in the paper. In the section “Sociological aspects” (line 409) only acceptance issues are analysed. I recommend the extension of this analysis also to digital divide. If literature does not cover this aspect, at least the authors should mention/reflect on this.
- Line 167: The authors mentioned “To our knowledge, this analysis has not been carried out yet”. It is important for the reader to understand the importance of the methodology framework and why the authors use such framework. The authors included in section 1 Introduction one paragraph that superficially refers to this (lines 146-154 on definition and lines 205-212 on the model).
Only one reference (reference 22) is provided to cover the definition and the model. This reference is, in my opinion, not applicable. Reference 22 refers to the description of the establishment of a sustainable network for HTA in Europe, as it can be read in the abstract: “This article presents the background, objectives, and organization of EUnetHTA, which involved a total of sixty-four partner organizations.”. The paper (reference 22) describes EUnetHTA, not the model that later the network developed.
A)
For the definition of HTA, the authors refer to Kristensen et al. (ref. 22). However, this definition is outdated. The correct reference to the definition of HTA should be:
O’Rourke, Brian, Wija Oortwijn, and Tara Schuller. 2020. “The New Definition of Health Technology Assessment: A Milestone in International Collaboration.” International Journal of Technology Assessment in Health Care 36 (3): 187–90. https://doi.org/10.1017/S0266462320000215.
(please see also: http://htaglossary.net/health-technology-assessment)
Since the authors choose to use HTA core model, not only the definition of HTA is important and the own definition of “health technology” is relevant (as also indicated in the new HTA definition, in Note 1. “A health technology is an intervention developed to prevent, diagnose or treat medical conditions; promote health; provide rehabilitation; or organize healthcare delivery. The intervention can be a test, device, medicine, vaccine, procedure, program, or system (definition from the HTA Glossary; http:// htaglossary.net/health+technology). “
By using a HTA framework, the authors are labelling the STR as a health technology. This, in my perspective, needs to be made explicit in the paper. Because this it indeed the novelty of the paper.
Otherwise, a methodology such as the one used in Technology Assessment (TA) could have also been used as a framework for the analysis (and there are several publications on the assessment of STR using and TA approach; see for instance publications by Michael Decker, Bettina Krings and Nora Weinberger).
B)
Regarding the Core model, correct references should be, for instance
Kristensen, Finn Børlum, Kristian Lampe, Claudia Wild, Marina Cerbo, Wim Goettsch, and Lidia Becla. 2017. “The HTA Core Model s — 10 Years of Developing an International Framework to Share Multidimensional Value Assessment.” Value in Health 20 (2): 244–50. https://doi.org/10.1016/j.jval.2016.12.010.
Or others as indicated in: https://eunethta.eu/hta-core-model/.
One aspect that caught my attention: The core model includes “Patients and social aspects” as one of the dimensions to be analysed. Are all the OAs analysed in the papers, patients?
- Line 356: The organizational aspects presented only focus on training of professionals and family members. I was wondering why the training of the user was not mentioned?
- Line 481: The way the paragraph is presented, refers to presentation of results. Thus, should be included in the previous section.
- Line 599: This paragraph should be presented in the Introduction section and its content used for comparison or reference in the discussion of results.
Minor comments:
- Line 49: is there a reason that “Consumer Science and Analytics Research” appear in bold?
- Line 109: should be increase instead of increased.
- Please check the consistency in the use of the abbreviation for Older Adults (OAs / OA).
- Line 138: it is mentioned “various authors” reported… However only two references are presented. I suggest being more precise with the language. Either mention “two authors” or keep the various but provide more references.
- Line 157: it is mentioned “(…) at home and institutions (…)”. Which kind of Institutions, does it refer to? Are the authors referring only to health and care institutions? Language should be more precise.
- Line 158-167: this part of the text refers to the methodology and should be included in section 2 Materials and Methods. It should not be in section 1. Introduction.
- Line 163-64: there is a typo. Repetition of wording.
- Please check the numeration in the Tables (Table 2, should be Table 1).
- Lines 199-200: the synthesis of the results should be presented in section 3 Results.
- Line 213: the flow chart, Figure 2, should be included at in Line 199, after “(…) implementation of the robots”.
- Line 208: in this paragraph the dimensions of the core model are identified. However, the legal aspects are missing.
- Line 209: there is a typo “technical technological”. “Technical” should be deleted.
- Figure 2. Please revise the typos (N= and n=) and in box “references identified by database search” replace et by and.
- Line 219: I suggest starting the sentence with “The analysed references…” instead of “All works”.
- Line 2019: after the first sentence, Table “Selected pertinent articles to the subject matter” should be introduced, as Table 1.
- Reference to Table “Studies description” is missing in the text. I suggest to included it in Line 223: After “In the selected publications (Figure 2) …”.
- Line 226: after [41], an “and” should be added.
- Line 301: this paragraph would benefit if more information were provided. Which benefits were reported? And what are the “other benefits”?
- Line 309: it seems that the contribution of the robot goes beyond the social sphere since they can assist “in diagnosis and follow-up”. This aspect could be a bit developed. How is the assistance made?
- Line 317-318: it seems to be a repetition, please confirm.
- Line 321: it is mention “reduce families workload”. In what sense? Related to travel times? Could be developed.
- Line 388: MCI – please write in full
- Line 389: HR please write in full
- Line 459: should it be 24 studies instead of 21 as written?
- Line 481: check repetition: mentioned showed.
Author Response
Answers to Reviewer 2
Overview and general recommendations:
The paper focus on social telepresence robots and their role when interacting with older adults, in relation to aspects of loneliness and social isolation. The authors choose a literature review, published in the last 20 year, thus including per and post COVID-19 time. The retrieved publications were analysed using a multidimensional framework, based on the HTA Core Model Dimensions.
Results of the paper are important and add knowledge to the current framing of assessment of social robots.
The main strength of the paper is the analysis conducted using an HTA approach and methodology. This is in my perspective, the novelty that the paper can bring to, not only the field of robotics but also to HTA field. However, the authors do not provide a justification why this method is in general important and even more important why it was selected to be used in the literature review, applied to social robots. Due to the technology under assessment, the reason for choosing this methodological framework is of great importance and the link to it is missing.
When reading the paper, I identify several issues that the paper should address, and it does not. Since this will imply more than a mere language check or small complements, I recommend that a major revision is necessary. However, I strongly encourage the authors to consider my suggestions, complement the paper and re-submit it.
Below, in more detail, I explain my perspective and concerns.
Major comments:
- Abstract: the reason why the review of literature was conducted is missing. The aim of the paper should be explicit for the reader, already in the abstract.
Answer: the reason why the review of literature was conducted was added in the abstract (please see abstract).
- Line 48 to 71: This paragraph refers exclusively to data from a questionnaire applied in France and no other country’s data is presented. Since the paper does not focus on a French perspective or is limited to an analysis in France, I was wondering why only such data was presented. Results from such questionnaires are important, but when presented in the paper, a short international perspective would be more interesting.
Answer: a short international perspective has been added as suggested by the referee (lines 87-92).
- In addition to the previous comment, a short indication of the total number of people over 60 exist in France would be useful so the reader can better frame the presented percentages.
Answer: a short indication of the total number of people over 60 exist in France as suggested by the referee (line 67)
- The paper focusses its analysis on older adults. An important factor related to this segment of the population is the “generational digital divide”, as mentioned in line 59. However, this is the only time that such important topic is approach in the paper. In the section “Sociological aspects” (line 409) only acceptance issues are analysed. I recommend the extension of this analysis also to digital divide. If literature does not cover this aspect, at least the authors should mention/reflect on this.
Answer: The analysis has been extended to digital divide which is now introduced in the results (lines 666_671) and discussion (lines 1136-1144).
- Line 167: The authors mentioned “To our knowledge, this analysis has not been carried out yet”. It is important for the reader to understand the importance of the methodology framework and why the authors use such framework. The authors included in section 1 Introduction one paragraph that superficially refers to this (lines 146-154 on definition and lines 205-212 on the model).
Answer: Explanations on the methodology framework were added (lines 222-234; lines 369-372; lines 418-436; table 1)
Only one reference (reference 22) is provided to cover the definition and the model. This reference is, in my opinion, not applicable. Reference 22 refers to the description of the establishment of a sustainable network for HTA in Europe, as it can be read in the abstract: “This article presents the background, objectives, and organization of EUnetHTA, which involved a total of sixty-four partner organizations.”. The paper (reference 22) describes EUnetHTA, not the model that later the network developed.
World Health Organization. (2011). Health technology assessment of medical devices. https://apps.who.int/iris/bitstream/handle/10665/44564/9789241501361-eng.pdf
Answer: This reference was added in the text (line 227) and in the references (ref 27)
A)For the definition of HTA, the authors refer to Kristensen et al. (ref. 22). However, this definition is outdated. The correct reference to the definition of HTA should be:
O’Rourke, Brian, Wija Oortwijn, and Tara Schuller. 2020. “The New Definition of Health Technology Assessment: A Milestone in International Collaboration.” International Journal of Technology Assessment in Health Care 36 (3): 187–90. https://doi.org/10.1017/S0266462320000215.
(please see also: http://htaglossary.net/health-technology-assessment)
Answer: The reference O’Rourke was added in the text (line 225) and in the references (ref 26)
Since the authors choose to use HTA core model, not only the definition of HTA is important and the own definition of “health technology” is relevant (as also indicated in the new HTA definition, in Note 1. “A health technology is an intervention developed to prevent, diagnose or treat medical conditions; promote health; provide rehabilitation; or organize healthcare delivery. The intervention can be a test, device, medicine, vaccine, procedure, program, or system (definition from the HTA Glossary; http:// htaglossary.net/health+technology). “
Answer: This definition was added (lines 185-194)
By using a HTA framework, the authors are labelling the MTR as a health technology. This, in my perspective, needs to be made explicit in the paper. Because this it indeed the novelty of the paper.
Answer: This link between MTR and “health technology” was made explicit (lines 185-194)
Otherwise, a methodology such as the one used in Technology Assessment (TA) could have also been used as a framework for the analysis (and there are several publications on the assessment of STR using and TA approach; see for instance publications by Michael Decker, Bettina Krings and Nora Weinberger).
B)Regarding the Core model, correct references should be, for instance
Kristensen, Finn Børlum, Kristian Lampe, Claudia Wild, Marina Cerbo, Wim Goettsch, and Lidia Becla. 2017. “The HTA Core Model s — 10 Years of Developing an International Framework to Share Multidimensional Value Assessment.” Value in Health 20 (2): 244–50. https://doi.org/10.1016/j.jval.2016.12.010.
Or others as indicated in: https://eunethta.eu/hta-core-model/.
One aspect that caught my attention: The core model includes “Patients and social aspects” as one of the dimensions to be analysed. Are all the OAs analysed in the papers, patients?
Answer: as shown in table 3, some OAs are patients but others are healthy persons
- Line 356: The organizational aspects presented only focus on training of professionals and family members. I was wondering why the training of the user was not mentioned?
Answer: The Older adults had a simple training. We have added a sentence to clarify this point(lines 702-704)..
- Line 481: The way the paragraph is presented, refers to presentation of results. Thus, should be included in the previous section.
Answer: as suggested by the referee, the paragraph has been included in the previous section (results) and the sentences have been rephrased. Lines 481-486 in the original manuscript are now in the results lines 743-747.
- Line 599: This paragraph should be presented in the Introduction section and its content used for comparison or reference in the discussion of results.
Answer: This paragraph has now been presented in the Introduction section (lines 212-215) and its content used for comparison in the discussion of results (lines 1173-1193)
Minor comments:
- Line 49: is there a reason that “Consumer Science and Analytics Research” appear in bold?
Answer: This has been rectified (line 65)
- Line 109: should be increase instead of increased.
Answer: This has been rectified.
- Please check the consistency in the use of the abbreviation for Older Adults (OAs / OA).
Answer: This has been rectified in all the text
- Line 138: it is mentioned “various authors” reported… However only two references are presented. I suggest being more precise with the language. Either mention “two authors” or keep the various but provide more references.
Answer: “Two authors” has now replaces various (line 205)
- Line 157: it is mentioned “(…) at home and institutions (…)”. Which kind of Institutions, does it refer to? Are the authors referring only to health and care institutions? Language should be more precise.
Answer: “health and care” has been added in the sentence (line 236)
- Line 158-167: this part of the text refers to the methodology and should be included in section 2 Materials and Methods. It should not be in section 1. Introduction.
Answer: this part of the text has been included in section 2 Materials and Methods (lines 362-372)
- Line 163-64: there is a typo. Repetition of wording.
Answer: This has been rectified
- Please check the numeration in the Tables (Table 2, should be Table 1).
Answer: the numeration in the table 1 and 2 have been changed
- Lines 199-200: the synthesis of the results should be presented in section 3 Results. ).
Answer: This has been rectified (line 590)
- Line 213: the flow chart, Figure 2, should be included at in Line 199, after “(…) implementation of the robots”.
Answer: the flow chart, Figure 2, has been included at in Line 398, after “(…) implementation of the robots”.
- Line 208: in this paragraph the dimensions of the core model are identified. However, the legal aspects are missing.
Answer: The Legal aspects have been added (line 434)
- Line 209: there is a typo “technical technological”. “Technical” should be deleted.
Answer: This has been rectified
- Figure 2. Please revise the typos (N= and n=) and in box “references identified by database search” replace et by and.
Answer: this has been rectified
- Line 219: I suggest starting the sentence with “The analysed references…” instead of “All works”.
Answer: this has been rectified according to the referee’s suggestion (line 525).
- Line 2019: after the first sentence, Table “Selected pertinent articles to the subject matter” should be introduced, as Table 1.
Answer: this has been rectified (line 528)
- Reference to Table “Studies description” is missing in the text. I suggest to included it in Line 223: After “In the selected publications (Figure 2) …”.
Answer: this has been rectified according to the referee’s suggestion (line 590).
- Line 226: after [41], an “and” should be added.
Answer: this has been rectified
- Line 301: this paragraph would benefit if more information were provided. Which benefits were reported?
Answer: We rephrased the sentence to clarify it. Indeed the benefit was a reduction of loneliness and isolation (lines 743-747).
And what are the “other benefits”?
Answer: details on the benefits for OAs, caregivers and professionals are provided in the paragraphs “Health-related quality of life and MTR use” (lines 747 -789) and “morbidity and MTR use” (lines 792-804).
- Line 309: it seems that the contribution of the robot goes beyond the social sphere since they can assist “in diagnosis and follow-up”. This aspect could be a bit developed. How is the assistance made?
Answer : as suggested by the referee, the “diagnosis and follow-up” has been developed in the paragraphe « morbidity and MTR use (lines 792-804).
- Line 317-318: it seems to be a repetition, please confirm.
Answer: we have removed the repetition
- Line 321: it is mention “reduce families workload”. In what sense? Related to travel times? Could be developed.
Answer: the sentence has now been developed in lines 763-767 : “Caregivers also considered that the MTR could contribute to improve OAs’ health, wellbeing, cognitive functioning, control of everyday activities such as medications and safety and thus reduce families’mental and physical workload [28]”.
- Line 388: MCI – please write in full:
Answer: this has been rectified (mild cognitive impairment)
- Line 389: HR please write in full
Answer: this has been rectified (heart rate)
- Line 459: should it be 24 studies instead of 21 as written?
Answer: this has been rectified (24 studies)
- Line 481: check repetition: mentioned showed. ?
Answer: the word ” showed” has been deleted
Reviewer 3 Report
The study has been done properly in all aspects. However, I do not see any interesting results that would inform the readers. On the other hand, it is essential to start work in this field. You could improve it by finding some interesting avenues to emphasize and proposing some future work.
Author Response
Answers to referee 3
You could improve it by finding some interesting avenues to emphasize and proposing some future work.
Answer : we have improved the manuscript by proposing some future work (lines 1193-1198).
Round 2
Reviewer 1 Report
I concur with the revised version.
Author Response
Thank you for your proofreading
Reviewer 2 Report
Dear Authors,
Thank you for taking into consideration my comments and for reviewing the manuscript accordingly and resubmitting it. In my perspective the article has improved. It is now clear that you are analysing MTR as health technologies, using an HTA framework.
I do have minor comments/suggestions, that I would ask for your consideration:
- In the abstract, please write in full older adults instead of OAs
- in section 3.2., please consider the use of Italic for the identification of the dimensions under analysis, to make the reading easier.
- small formation issues need to be corrected; Lines need to be added in section 3.2.7 for instance and check paragraphs. not all of them contain the full stop sign.
Author Response
All new changes have been highlighted in green to facilitate their identification within the manuscript
Answers to Reviewer 2:
Dear Authors,
Thank you for taking into consideration my comments and for reviewing the manuscript accordingly and resubmitting it. In my perspective the article has improved. It is now clear that you are analysing MTR as health technologies, using an HTA framework.
I do have minor comments/suggestions, that I would ask for your consideration:
- In the abstract, please write in full older adults instead of OAs
Answers:
We have replaced OAs by "older adults" in the abstract.
- in section 3.2., please consider the use of Italic for the identification of the dimensions under analysis, to make the reading easier.
Answers:
We have italicised the dimensions analysed for ease of reading
- small formation issues need to be corrected; Lines need to be added in section 3.2.7 for instance and check paragraphs. not all of them contain the full stop sign.
Answers:
Additional lines have been added to make the paragraph easier to read.
Missing full stops at the end of each sentence have been added.
